# Ascertaining provider-level implicit bias in electronic health records with rules-based natural language processing: A pilot study in the case of prostate cancer

**Ashwin Ramaswamy**[1], **Michael Hung**[1], **Joe Pelt**[1], **Parsa Iranmahboub**[1], **Lina P. Calderon**[1], **Ian S. Scherr**[1], **Gerald Wang**[1], **David Green**[1], **Neal Patel**[1], **Timothy D. McClure**[1], **Christopher Barbieri**[1], **Jim C. Hu**[1], **Charlotta Lindvall**[2,3], **Douglas S. Scherr**[1] *

1 Department of Urology, NewYork-Presbyterian Hospital/Weill Cornell Medical Center, New York, New York, United States of America, 2 Dana Farber Cancer Center, Boston, Massachusetts, United States of America, 3 Harvard Medical School, Boston, Massachusetts, United States of America

* dss2001@med.cornell.edu

**Data Availability Statement:** The EHR datasets cannot be shared for the privacy of the individuals whose data were used in this study given it is entirely populated with protected health

## Abstract

### Purpose

Implicit, unconscious biases in medicine are personal attitudes about race, ethnicity, gender, and other characteristics that may lead to discriminatory patterns of care. However, there is no consensus on whether implicit bias represents a true predictor of differential care given an absence of real-world studies. We conducted the first real-world pilot study of provider implicit bias by evaluating treatment parity in prostate cancer using unstructured data–the most common way providers document granular details of the patient encounter.

### Methods and findings

Patients $\geq$18 years with a diagnosis of very-low to favorable intermediate-risk prostate cancer followed by 3 urologic oncologists from 2010 through 2021. The race Implicit Association Test was administered to all providers. Natural language processing screened human annotation using validated regex ontologies evaluated each provider's care on four prostate cancer quality indicators: (1) active surveillance utilization; (2) molecular biomarker discussion; (3) urinary function evaluation; and (4) sexual function evaluation. The chi-squared test and phi coefficient were utilized to respectively measure the statistical significance and the strength of association between race and four quality indicators. 1,094 patients were included. While Providers A and B demonstrated no preference on the race Implicit Association Test, Provider C showed preference for White patients. Provider C recommended active surveillance (p<0.01, φ = 0.175) and considered biomarkers (p = 0.047, φ = 0.127) more often in White men than expected, suggestive of treatment imparity. Provider A considered biomarkers (p<0.01, φ = 0.179) more often in White men than expected. Provider B demonstrated treatment parity in all evaluated quality indicators (p>0.05).

information which can be used to identify specific individuals, even after our efforts to anonymize patient name, medical record number, and date of birth. Human research participant data includes: names, location data, date of birth, examination dates, sex, ethnicity, and race. Data sets may be available (with limitations) on request through Weill Cornell Medicine Human Research Protections at 575 Lexington Avenue, New York, NY 10022, Phone: (646)-962-8200.

**Funding:** The author(s) received no specific funding for this work.

**Competing interests:** NO authors have competing interests.

## Conclusions

In this pilot study, providers' practice patterns were associated with both patient race and implicit racial preferences in prostate cancer. Alerting providers of existing implicit bias may restore parity, however future assessments are needed to validate this concept.

## Introduction

Implicit biases in medicine are personal attitudes or beliefs about race, ethnicity, sex, disability, and other characteristics that may unconsciously influence clinical decision making and lead to discriminatory patterns of care. Though rarely harboring overtly explicit prejudices, the majority of providers hold implicit bias against marginalized groups [1–3]. These stereotyped beliefs, which discriminate against those already vulnerable to their health conditions, are learned over time through cultural exposure to the modern healthcare environment and are notoriously difficult to change [4, 5]. This is especially troubling as there is widespread evidence reaffirming a clear association between minority group status and negative clinical outcomes across medicine [6–8]. Though rarely harboring overtly explicit prejudices, most providers hold implicit bias against marginalized groups that are notoriously difficult to change [1]. These stereotyped beliefs are especially troubling given widespread evidence associating minority group status with negative clinical outcomes across medicine.

However, subsequent studies have not reached consensus on whether implicit bias represents a true predictor of differential care. While some series demonstrate that provider implicit bias independently influences care delivery [9], others argue that "physicians' implicit bias does not impact clinical decision making" [10]. Importantly, this unclear relationship is belied by the near-total absence of studies using real-world data [11].

Unstructured clinical narratives, which make up nearly 80% of clinical information in the electronic health record (EHR) [12], represents a unique opportunity to capture implicit bias as it represents the most convenient way providers document granular details of clinical management not well-populated in discrete datasets [13]. Applying recent advances in natural language processing (NLP) to real-world EHRs [14], we explored the relationship between providers' implicit bias and their respective treatment parity in prostate cancer, a condition associated with guideline-based care [15], validated functional outcomes, and well-documented disparities in care [16–18].

## Methods

### Study design

This was a retrospective, observational pilot study of new patients ≥18 years presenting with an elevated PSA or prostate cancer to all three urologic oncologists (average 17 years in practice, average $h$-index of 51) affiliated with Weill Cornell Medical Center from 2010 to 2021. All note-level data was extracted from the EHR (Epic®, Verona, WI) including new and follow-up visit notes, as well as telephone encounters, MyChart messages, pathology reports, and radiology reports. Notes were associated with patient age, race, ethnicity, body mass index (BMI), laboratory values, and procedure history. Patient race was categorized as White or Not White as determined by patient-reported designations on clinical intake forms. Patients were excluded if they did not have an associated prostate cancer diagnosis or a missing PSA associated with the first encounter.

## Population

We decided to study men with very low-risk or low-risk prostate cancer and men with favorable intermediate-risk prostate cancer, as specified by the National Comprehensive Cancer Network (NCCN) Prostate Cancer Guidelines [19]. These guidelines stratify risk based on histologic grade group (GG), which ranges from 1 (most favorable) to 5 (least favorable), PSA, tumor volume, and other associated measures. These men are considered to be highly appropriate for active surveillance (AS) by most clinicians [20] and is the preferred management for very low-risk or low-risk prostate cancer as recommended by the American Urologic Association, American Society for Radiation Oncology, Society of Urologic Oncology, and NCCN, and is a viable option for men with favorable intermediate-risk prostate cancer with a PSA < 10 or low volume disease (defined as < 50% positive biopsy cores) [19, 21, 22]. For the purposes of this study, we categorized any man with GG1 prostate cancer & a visit-matched PSA < 10 to be low-risk (including very low-risk), and any man with GG1 prostate cancer & a visit-matched PSA > 10 *or* GG2 prostate cancer with < 50% positive cores to be favorable intermediate-risk.

## Outcomes

Four quality indicators reflecting guideline-concordant [19, 21] prostate cancer care–which are uniquely [13] and accurately [23] captured in EHR unstructured clinical narratives–were chosen to evaluate each provider's clinical decision making in the aforementioned patient cohort: (1) utilization of AS, (2) discussion of prognostic (e.g., Decipher [24]) prostate cancer biomarkers, use of which is associated with improved risk stratification [25], (3) discussion of the patients' urinary function, and (4) discussion of the patients' sexual (i.e., erectile) function [21]. This information has historically been inaccessible in the literature, given the laborious process of manual chart review required to extract discrete and usable data elements.

Natural language processing (NLP), here defined as work that computationally represents, transforms, or utilizes text, was employed to iteratively identify provider performance in the four quality indicators of prostate cancer care across each patient's unstructured clinical narrative documentation. Specifically, we employed NLP-screened human annotation to ascertain provider performance in the four quality indicators for each patient using validated regular expressions (regex) ontologies to display only the relevant outcomes-in-context enabling considerably expedited data abstraction [14]. Ontology development erred towards improved *recall* to maximize retrieval of relevant positive "hits", as "misses" would not be reviewed. Final ontologies included various potential abbreviations, word phrasing, and punctuation; furthermore, utilizing regex enabled us to create context-dependent rules, such as excluding those notes in which the text "`proceed with prostatectomy`" either preceded or succeeded the text "`active surveillance` The four quality outcomes were measured in all unstructured data from the first visit to 1 year thereafter. Provider-specific dot phrases were excluded to prevent automated text from being considered as implemented care. An in-depth description of regex ontologies, validation, and the software used is in **S1 Appendix**.

Lastly, the race Implicit Association Test (IAT) was administered to all providers to assess implicit bias for White vs. Black people's faces. IAT results were administered after data collection was performed. The IAT is the most widely used test to evaluate unconscious preferences across a spectrum of social categories [26]. The IAT is a series of computerized categorization tasks in which participants sort stimuli into opposing categories as quickly and accurately as possible by rapidly pressing "I" and "E" keys; in the race IAT, the participant is tasked with sorting pictures of White or Black people's faces with words with positive (e.g., smile, diamond) or negative (e.g., grief, crash) connotation presented in a random order. The test is

predicated on the assumption that the ease with which an individual sorts concepts together is indicative of implicit bias. The IAT is scored on a 7-point relative preference measure of strong, moderate, slight, preference for either race with no racial preference set in the middle. IAT results were not known to the providers prior to data collection.

## Statistical analysis

Baseline patient characteristics were assessed using descriptive patient-level statistics; the Mann-Whitney U test and chi-squared test of independence were used to compare continuous and categorical variables, respectively, at an alpha of 0.05. We calculated a phi coefficient ($\varphi$) to measure the strength of association between race and the quality indicators and to enable standardized comparison of preferential treatment of White men between the three providers.

All analyses were anonymized to patient name, medical record number, location data, and date of birth prior to access on March 13, 2023; study authors did not have access to de-anonymized data during nor after analysis was performed. Statistical analysis was performed in R 3.4.1 (R Foundation for Statistical Computing, Vienna, Austria). All analyses adhered to STROBE guidelines and were approved by the Weill Cornell Medicine Institutional Review Board (#22–03024547); the requirement for informed consent was waived as our study met the criteria of a minimal risk study under 45 CFR 46.116(f). All authors declare no financial or non-financial competing interests.

## Results

We identified 5,981 (associated with 211,345 data entry points; 119,438 unstructured, 91,907 structured) patients who were diagnosed with elevated PSA or prostate cancer from January 2010 –December 2021. Of the 1,758 patients who had complete data for PSA, race/ethnicity, and prostate biopsy pathology, 1,094 were identified as having very low, low, or favorable intermediate risk prostate cancer per NCCN risk stratification and were included for analysis: of this sample, 67% and 33% of patients were White and non-White, respectively. Additionally, while Providers A and B demonstrated no racial preference on the race IAT, Provider C had a slight preference for White patients.

In aggregate across all providers, (1) 31.0% patients were recommended to initiate or continue AS, (2) molecular biomarkers were considered in 27.2% patients, (3) urinary function was evaluated in 58.1% patients, and (4) erectile function was evaluated in 56.4% patients (**Table 1**). Older age was significantly associated with providers' decision to initiate or continue AS (median [IQR], 67 [61–72] vs. 64 [58–69] years, p<0.001), evaluate urinary function (66 [59–70.5] vs. 64 [58–70] years, p = 0.023), and not evaluate erectile function (67 [60–72] vs. 64 [58–69], p<0.001). Molecular biomarker discussion (p<0.001) and urinary function evaluation (p = 0.033) occurred more often in White men than expected, suggestive of treatment imparity. Neither BMI, baseline PSA, nor income were associated with the quality indicators.

Provider A considered or utilized molecular biomarkers (p = 0.001, $\varphi$ = 0.179) more often in White men than expected (**Table 2 and Fig 1**). Provider C recommended initiating or continuing AS (p = 0.006, $\varphi$ = 0.175) and considered molecular biomarkers (p = 0.046, $\varphi$ = 0.127) more often in White men than expected. Provider B demonstrated treatment parity in all evaluated quality indicators (p>0.05). All providers demonstrated treatment parity in urinary and erectile function evaluation (p>0.05).

## Discussion

In this novel study utilizing NLP in the evaluation of unstructured EHRs, we found significant racial variation in the management of men with prostate cancer. In the cumulative delivery of

**Table 1. Quality Indicators for men with very low to favorable intermediate risk prostate cancer.**

| Quality Indicators | Active Surveillance (AS) | | | Molecular Biomarkers | | | Discussion of Urinary Function (UF) | | | Discussion of Erectile Function (EF) | | |
|---|---|---|---|---|---|---|---|---|---|---|---|---|
| | AS–No | AS–Yes | P | Biomarker–No | Biomarker–Yes | p | UF–No | UF–Yes | p | EF–No | EF–Yes | p |
| No. | 898 | 401 | | 946 | 353 | | 544 | 755 | | 567 | 732 | |
| Age, years (median (IQR)) | 64.0 [58.0, 69.0] | 67.0 [61.0, 72.0] | <**0.001**\* | 65.0 [59.0, 70.0] | 65.0 [59.0, 70.0] | 0.737 | 64.0 [58.0, 70.0] | 66.0 [59.0, 70.5] | **0.023**\* | 67.0 [60.0, 72.0] | 64.0 [58.0, 69.0] | <**0.001**\* |
| BMI (median (IQR)) | 26.6 [24.4, 29.3] | 26.1 [24.3, 28.7] | 0.086 | 26.5 [24.4, 29.3] | 26.3 [24.4, 28.7] | 0.723 | 26.5 [24.4, 29.2] | 26.5 [24.4, 29.1] | 0.895 | 26.4 [24.4, 29.2] | 26.5 [24.4, 29.0] | 0.962 |
| PSA ng/mL (median (IQR)) | 5.1 [3.7, 7.0] | 5.2 [3.9, 7.2] | 0.411 | 5.1 [3.7, 7.0] | 5.3 [4.0, 7.4] | 0.051 | 5.2 [3.9, 7.5] | 5.1 [3.7, 6.9] | 0.090 | 5.1 [3.8, 7.2] | 5.2 [3.8, 7.0] | 0.907 |
| Race Group | | | 0.067 | | | <**0.001**\* | | | **0.033**\* | | | 0.150 |
| White | 584 (65.0) | 287 (71.6) | | 605 (64.0) | 266 (75.4) | | 343 (63.1) | 528 (69.9) | | 364 (64.2) | 507 (69.3) | |
| Not White | 198 (22.0) | 73 (18.2) | | 221 (23.4) | 50 (14.2) | | 128 (23.5) | 143 (18.9) | | 130 (22.9) | 141 (19.3) | |
| Unknown | 116 (12.9) | 41 (10.2) | | 120 (12.7) | 37 (10.5) | | 73 (13.4) | 84 (11.1) | | 73 (12.9) | 84 (11.5) | |
| Income Group | | | 0.092 | | | 0.213 | | | 0.508 | | | 0.495 |
| < 60k | 121 (13.8) | 41 (10.4) | | 127 (13.7) | 35 (10.2) | | 66 (12.5) | 96 (12.9) | | 75 (13.6) | 87 (12.1) | |
| 60k - 200k | 712 (81.4) | 341 (86.3) | | 761 (82.2) | 292 (84.9) | | 435 (82.4) | 618 (83.3) | | 451 (81.6) | 602 (84.0) | |
| >200K | 42 (4.8) | 13 (3.3) | | 38 (4.1) | 17 (4.9) | | 27 (5.1) | 28 (3.8) | | 27 (4.9) | 28 (3.9) | |

Numbers are No. (%) unless otherwise noted. SD = standard deviation, IQR = interquartile range

guideline-concordant urologic care, providers considered molecular biomarkers and evaluated urinary function more often in White men than expected; in contrast, providers demonstrated racial parity in both utilizing AS and evaluating erectile function. However, our analysis of granular unstructured data revealed provider-level racial imparity in clinical decision making which both *corroborates* and *belies* clinic-level observations. For example, while Provider A & C's molecular biomarker imparity *corroborates* observed clinic-level imparity, Provider C's AS

**Table 2. Contingency table of prostate cancer quality indicators by race, stratified by provider.**

| (Race IAT) | All Providers | | | Provider A | | | Provider B | | | Provider C | | |
|---|---|---|---|---|---|---|---|---|---|---|---|---|
| | | | | (No preference) | | | (No preference) | | | (Slight White preference) | | |
| | White | Not White | p | White | Not White | p | White | Not White | p | White | Not White | p |
| No. | 871 | 271 | - | 290 | 122 | - | 366 | 70 | - | 193 | 53 | - |
| **(1) Active Surveillance** | | | 0.063 | | | 0.690 | | | 0.857 | | | **0.006**\* |
| *Observed* | 287 (33.0) | 73 (26.9) | | 101 (34.8) | 40 (32.8) | | 82 (22.4) | 15 (21.4) | | 103 (53.4) | 17 (32.1) | |
| *Expected* | 274.6 (31.5) | 85.4 (31.5) | | 99.3 (34.2) | 41.8 (34.2) | | 81.4 (22.3) | 15.6 (22.3) | | 94.2 (48.8) | 25.9 (48.8) | |
| **(2) Molecular Biomarkers** | | | **0.001**\* | | | **0.001**\* | | | 0.333 | | | **0.046**\* |
| *Observed* | 266 (30.5) | 50 (18.5) | | 113 (39.0) | 25 (20.5) | | 76 (20.8) | 11 (15.7) | | 76 (39.4) | 13 (24.5) | |
| *Expected* | 241.0 (27.7) | 75.0 (27.7) | | 97.1 (33.5) | 40.9 (33.5) | | 73.0 (20.0) | 14.0 (20.0) | | 69.8 (36.2) | 19.2 (36.2) | |
| **(3) Urinary Function** | | | **0.022**\* | | | 0.772 | | | 0.551 | | | 0.127 |
| *Observed* | 528 (60.6) | 143 (52.8) | | 97 (33.4) | 39 (32.0) | | 259 (70.8) | 52 (74.3) | | 163 (84.5) | 40 (75.5) | |
| *Expected* | 511.8 (58.8) | 159.2 (58.8) | | 95.7 (33.0) | 40.3 (33.0) | | 261.0 (71.3) | 49.9 (71.3) | | 159.3 (82.5) | 43.7 (82.5) | |
| **(4) Erectile Function** | | | 0.073 | | | 0.928 | | | 0.824 | | | 0.610 |
| *Observed* | 507 (58.2) | 141 (52.0) | | 125 (43.1) | 52 (42.6) | | 246 (67.2) | 48 (68.6) | | 131 (67.8) | 34 (64.2) | |
| *Expected* | 494.2 (56.7) | 153.8 (56.7) | | 124.6 (43.0) | 52.4 (43.0) | | 246.8 (67.4) | 47.2 (67.4) | | 129.5 (67.1) | 35.7 (67.1) | |

Numbers are No. (%) unless otherwise noted. SD = standard deviation, IQR = interquartile range

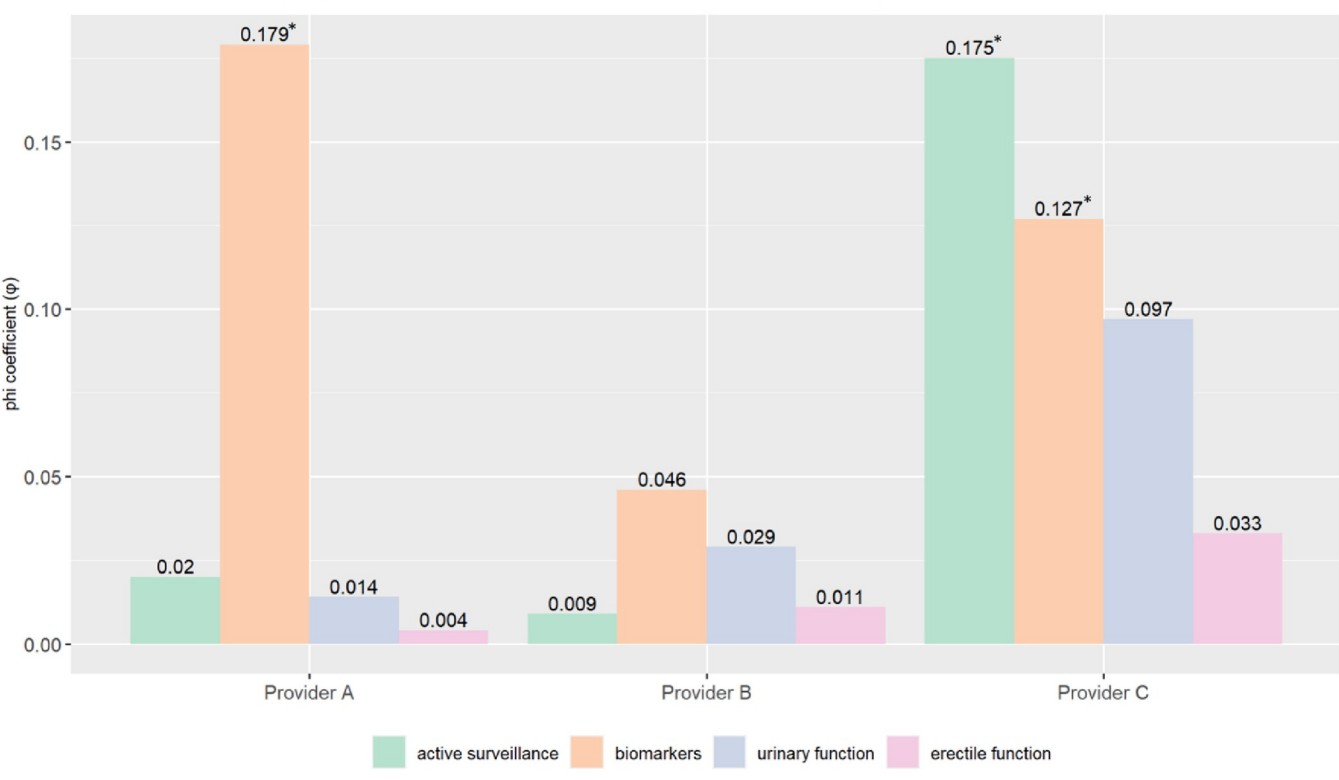

**Fig 1. Degree of association between provider quality indicators (active surveillance utilization [green], molecular biomarker discussion [orange], urinary function evaluation [blue], and sexual function evaluation [pink]) and race among each provider evaluated via the phi coefficient with the asterisks indicating statistical significance at an alpha of 0.05.**

imparity *belies* our finding that White and non-White men receive care equity in aggregate. Notably, unlike previous studies using clinical vignettes, IAT results imperfectly correlate with provider treatment parity.

Despite the best intentions by providers and health systems to ensure the delivery of equitable care, disparities in healthcare outcomes exist in nearly every disease even after controlling for factors such as access to care, socioeconomic status, or insurance coverage [27]. This determination, made in the Institute of Medicine (IOM)'s *Unequal Treatment* 2002 Report to Congress, is suggested to be caused in part by "bias, stereotyping, and prejudice on the part of healthcare providers", although direct evidence supporting this association was considered lacking at the time of the report's publication [27]. In ensuing decades, researchers continued to call attention to the pervasiveness of unequal care in a broad range of medical disciplines, while concurrently demonstrating that the majority of healthcare providers *do* harbor some degree of implicit bias [1–3]. However, the abundant use of hypothetical clinical vignettes and the near-total absence of real-world data in implicit bias studies has yielded only circumstantial–and conflicting–evidence regarding the possible relationship between provider implicit bias and observed disparities in patient outcomes [28, 29].

The challenges of using real-world patient data to evaluate implicit bias is limited by an imperfect methodology by which to do so. For example, the only study utilizing actual health outcomes inferred provider racial preferences in hypertensive treatment intensification solely

on the basis of pharmacy refills, a poor proxy of clinician intent [10, 30]. Furthermore, not all disparities between patient populations are necessarily discriminatory. To attribute a difference to a disparity requires a granular understanding of clinical context which can be uniquely characterized by unstructured EHR data [31]. Use of these records enables the reliable interpretation of a provider's clinical decision-making as tailored to a specific disease profile across an individual patient's trajectory through a healthcare system [23, 32]. Extracting usable data elements from clinical narratives has traditionally been infeasible given the laborious process of "gold standard" manual chart review. However, as demonstrated in this study, rules-based NLP-assisted human annotation enables the efficient abstraction of EHR outcomes that is broadly validated [33–35], widely generalizable to a range of clinical contexts [14, 36], demands minimal investment in computing resources and upfront cost [37], and not limited by non-interrogable "black box" models common in machine learning (ML) algorithms [38]. Compared to ML, rules-based algorithms better suited to evaluating treatment parity using EHR data: they can flexibly be adapted to the interpretation of unformatted "messy" data in small populations in which significant domain knowledge, high precision, and transparency are required. Furthermore, rules-based methodologies as described are foundational in a future of automated, accurate interpretation of healthcare data.

Prostate cancer is a well-suited candidate condition to evaluate the association of racial implicit bias and provider treatment parity using unstructured EHR data. First, there are increasingly apparent racial disparities in prostate cancer care–lower rates of AS [39–42], higher rates of treatment regret [43], higher cancer-specific mortality [44]–which are broadly attributable to social, rather than biologic, determinants of health. Second, prostate cancer only affects patients with male anatomy, thereby broadly removing sex as a confounding study variable (the notable exception being intersex or transgender patients). Third, prostate cancer screening is generally considered only within a narrow age range (approximately 50–69 years) with guidelines discouraging routine testing in older men, thereby largely removing age as a confounding study variable [45]. Lastly, the quality of prostate cancer care is measurable by outcomes found sparingly in structured data sources, such as whether a provider assessed a patients' baseline-disease specific function (e.g., urinary, sexual function) [21]. However, these outcomes are richly populated in unstructured clinical documentation, which effectively make existing structured data sources–either claims or registry–incompletely represent prostate cancer quality [46] and altogether inadequate at explaining observed *provider-level* deviations from guideline-concordant care [47].

Our study has important implications for clinical practice. First, by utilizing a previous validated methodology using NLP on unstructured EHR data, we conducted the first study of implicit bias to observe providers in their real work environments, without which one cannot understand how implicit bias translates into differential patient care [28]. Our novel approach enables a means to granularly measure healthcare disparities at the provider-level that could be integrated into routine clinical documentation inquiry and hospital reporting frameworks. Furthermore, unlike automated deep-learning models, our outlined process of NLP-screened human annotation is uncomplicated, customizable, interrogable, and–perhaps most importantly–utilizes supervised "human-in-the-loop" determinations. Second, our results naturally suggest clinician-facing, data-driven interventions to improve awareness of implicit bias. Provider awareness of implicit bias has been shown to encourage more egalitarian patterns of care [9]. Similar clinician-facing prompting interventions have been demonstrated to effectively promote behavior change, particularly among minoritized patients [48]. Notably, no current bias-reducing efforts, ranging from mental-debiasing interventions to group-administered trainings, have been demonstrated to reduce implicit bias [31, 49]. The approach outlined by our study may help healthcare systems periodically identify potentially clinically significant

provider-level disparities in care, tailor personalized prompting interventions, further evaluate the effect of race-concordant care, as well as measure the success of any corrective efforts to reduce discriminatory disparities.

Our study has several limitations. First, the NLP model was developed at a single urban, tertiary-care, academic institution among three providers and should be viewed as a pilot study; generalizability to more providers and other health systems may require a separate validation process. Second, treatment algorithms are not absolute; shared decision making plays an integral role in prostate cancer management where the decision to pursue AS may be highly personalized. Additionally, there are patient factors that may influence treatment that are not captured by our analysis including a family history of malignancy, anxiety about a cancer diagnosis, performance status, and medical comorbidities. To limit these confounders, our analysis included only men with very low to favorable intermediate risk prostate cancer for whom AS is a recommended option by guidelines. Third, we acknowledge that our NLP method relies on accurate documentation during a patient encounter. Failure to explicitly document the contents of shared decision making or findings such as normal urinary and erectile function despite assessments made during an encounter cannot be interpreted by our NLP method. However, this is a limitation of EHR documentation that exists even with manual chart review. Fourth, the IAT results were administered at one point in time and may not be a predictor of care across the entire time interval of our data. Similarly, we were not able to adjust for year of counseling as provider practices may have changed over time. Fifth, use of racial categories are artificial, social constructs that may poorly reflect the experiences of the categorized people; our choice to compare "White" and "non-White" was predicated on a recommendation to understand race not as a measure of biologic difference, but rather as a proxy for exposure to racism [50], which in the United States has historically been driven by White supremacy [51]. Additionally, there is added challenges in racial categorization based on EHR documentation, which further encouraged us to avoid extensive disaggregation of racial designations [52]. Lastly, while we were able to rule out age, weight, and income as potential confounders, we were unable to account for a range of additional socioeconomic factors that may contribute to treatment imparity, such as disability, education, or religion.

## Conclusions

In this pilot study, overall clinical practice patterns were associated with both patient race and implicit provider racial preferences among urologic oncologists in the management of prostate cancer. Further studies involving more quality indicators and providers may be warranted. Alerting providers of existing implicit bias can restore parity in clinical care but future assessments are needed to validate this concept.

## Supporting information

**S1 Appendix.**
(DOCX)

## Author Contributions

**Conceptualization:** Ashwin Ramaswamy, Michael Hung, Douglas S. Scherr.

**Data curation:** Ashwin Ramaswamy, Michael Hung, Joe Pelt, Parsa Iranmahboub, Lina P. Calderon, Ian S. Scherr.

**Formal analysis:** Ashwin Ramaswamy, Michael Hung, Joe Pelt, Parsa Iranmahboub, Lina P. Calderon, Douglas S. Scherr.

**Investigation:** Ashwin Ramaswamy, Michael Hung, Parsa Iranmahboub, Ian S. Scherr, Douglas S. Scherr.

**Methodology:** Ashwin Ramaswamy, Michael Hung, Joe Pelt, Parsa Iranmahboub, Charlotta Lindvall, Douglas S. Scherr.

**Project administration:** Douglas S. Scherr.

**Resources:** Ashwin Ramaswamy, Michael Hung, Joe Pelt, Gerald Wang, David Green, Neal Patel, Timothy D. McClure, Christopher Barbieri, Jim C. Hu, Douglas S. Scherr.

**Software:** Ashwin Ramaswamy, Michael Hung, Joe Pelt, Parsa Iranmahboub, Charlotta Lindvall.

**Supervision:** Douglas S. Scherr.

**Validation:** Ashwin Ramaswamy, Michael Hung, Joe Pelt, Parsa Iranmahboub, Lina P. Calderon, Charlotta Lindvall.

**Visualization:** Ashwin Ramaswamy, Michael Hung.

**Writing – original draft:** Ashwin Ramaswamy, Michael Hung.

**Writing – review & editing:** Ashwin Ramaswamy, Michael Hung, Joe Pelt, Parsa Iranmahboub, Lina P. Calderon, Gerald Wang, David Green, Neal Patel, Timothy D. McClure, Christopher Barbieri, Jim C. Hu, Charlotta Lindvall, Douglas S. Scherr.

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
