## [Decision Letter · Decision Letter 0]

17 Sep 2024

PONE-D-24-33113Ascertaining provider-level implicit bias in electronic health records with rules-based natural language processing: a pilot study in the case of prostate cancerPLOS ONE

Dear Dr. Ramaswamy,

Thank you for submitting your manuscript to PLOS ONE. After careful consideration, we feel that it has merit but does not fully meet PLOS ONE’s publication criteria as it currently stands. Therefore, we invite you to submit a revised version of the manuscript that addresses the points raised during the review process.

We look forward to receiving your revised manuscript.

Kind regards,

Alvaro Galli

Academic Editor

PLOS ONE

Journal Requirements:

Reviewers' comments:

Reviewer's Responses to Questions

**Comments to the Author**

1. Is the manuscript technically sound, and do the data support the conclusions?

Reviewer #1: Yes

Reviewer #2: Yes

2. Has the statistical analysis been performed appropriately and rigorously? 

Reviewer #1: Yes

Reviewer #2: Yes

3. Have the authors made all data underlying the findings in their manuscript fully available?

Reviewer #1: Yes

Reviewer #2: No

4. Is the manuscript presented in an intelligible fashion and written in standard English?

Reviewer #1: Yes

Reviewer #2: Yes

5. Review Comments to the Author

Reviewer #1: The authors have submitted a thoughtful and compelling study. It is novel, thoughtful, and merits publication.

The authors examined whether implicit biases among providers affect treatment decisions in prostate cancer care using natural language processing analysis of medical records. Using case records of three urologic oncologists to over 1,000 patients, they analyzed care quality and intensity in four domains: active surveillance, molecular biomarker discussion, and evaluations of urinary and sexual function. The providers took an Implicit Association Test to measure racial biases. The authors offer that the implicit association test results showed that two of the providers did not have implicit bias favoring White patients while one did. The 4 care criteria provided differed for two of the providers in statistically significant ways, one of which was the provider with implicit bias. These findings again emphasize that CaP care may reflect aspects of the provider rather than the patients and that implicit biases might be included as such a factor. While IAT testing has gained in its use, most have recognized that the results reflect beliefs rather than actions. This paper adds an important perspective by looking at such actions (care provided).

The authors appropriately argue that these results are appropriately a pilot study and highlights the need for further research- as well as potential interventions- to address biases in care.

I offer suggestions, both major and minor, that might be helpful to the manuscript.

Major:

1. Three providers: The N for the study is really 3 providers rather than 1000+ patients.

a. Recommend that this be expressed as the first limitation in discussion rather than the fourth.

2. IAT raw scores: If we are speaking about the same IAT, the test generates scores on a continuous scale that are subsequently categorized (-3 to 3)

a. The raw scores may be informative for Providers A and B, even if they are in the 0-0.15 range

b. The authors should state specifically that these numeric scores were either negative (a Black implicit bias) or positive numbers (a White implicit bias)

3. Non-White versus Black: The relationship between the independent variable IAT testing (White/Black bias) as a predictor of the dependent variable- patient-by-patient care provided as analyzed by NLP analysis are not matched (White/Non-White).

a. In addition to those self-identifying as Black, exactly who is non-White? Can this be further broken down?

i. Can the analysis also be run for just patients who have identified as Black and White?

b. The breakdown of populations into “white” and “non-white” has a troublesome history (one-drop theory, social hierarchy, justification for slavery, etc.).

i. I do not believe that this is what the authors are implying, and a brief description of what is NOT meant could be in the limitations.

Minor:

1. Self-identified race of the providers: We learn some facts about the 3 providers, but not about their race

a. We learn years of experience and a metric of publication impact (h-factor), which perhaps may have been included to suggest that experienced scientists cannot be biased. This potentially can be included in the limitations section.

b. Interestingly for a publication about race and bias, we do not learn the self-identified race of the individuals. This is not meant to assign an explanation for their bias, but it allows a very brief mention in the discussion as to who mainly provides care to Black patients (and to point out that having more providers- and a diverse sample-in future studies allows for a deeper inquiry of racial concordant/racial discordant care).

2. SES variables: We have income, but not education. Educational attainment of the patients could be important (bias against low education, educational attainment being a confounder)

a. Education might be an important variable. If not available for analysis, that should be indicated as a limitation.

3. More about GG: GG1 and GG2 could be more explicitly defined as this article may be read widely and outside of urological circles

4. IAT results: More information needed.

a. Knowing that the IAT results were not known to the providers would be important

b. Likely the IAT test was done once with each provider. At what point was this done? That bias at one testing session is a predictor for care over 12+ years is a difficult issue and simply can be discussed as a limitation.

i. It may have been done late in the timeframe of the cohort, meaning that one’s attitude in 2020 was used to predict care delivered in 2010

ii. Bias may not be constant over time (we hope it gets better!)

5. NLP will likely not be able to verify to what extent shared decision making actually took place. We're relying on what the provider included in the note to represent this. This is a limitation.

6. Getting a PSA test is also subject to bias: Everyone in this study already has a “positive” PSA value. That every patient getting into this analysis passed that barrier might suggest that the effect of bias in CaP care is likely larger than just these 4 criteria used (i.e., the need to consider the entire cascade of care framework).

Reviewer #2: Ramaswamy et al. present data on implicit bias evaluation in prostate cancer patients using natural language processing. I´ve really enjoyed reading this manuscript and the authors should be congratulated for this really good use NLP. However, sample size (3 providers, 1000 patients) and granularity of data is limited.

Major issue #1:

One of the significant findings was that " Provider C recommended active surveillance (p<0.01, �=0.175) and considered biomarkers (p=0.047, �=0.127) more often in White men than expected, suggestive of treatment imparity". I would consider caution on that conclusion. Afro-Americans are often diagnosed at younger age and have more aggressive disease. I would recommend to re-run the analysis applying a more granular approach on race/ethnicity (Caucasian, Hispanics, Afro-American, Others).

Major issue #2:

Because MRI fusion biopsy cause artificial stage migration to some degree, I was wondering, if the proportion of fusion biopsy was equal in the different population or if this was considered during analysis. There are some works, finding lower rates of MRI fusion biopsy in disadvantaged population. Because active surveillance is more safe in an MRI fusion biopsy pathway, AS is more frequently recommended if a fusion biopsy was performed. This could have introduced a systematic bias.

Major issue #3:

Did you adjust the analysis for the year of counselling? (biomarker and AS were more implemented in recent years) Did you adjust the analysis on socioeconomic status and insurance coverage of biomarkers etc?

Minor issue #1:

I think it should be mentioned in the limitations that all providers were highly proliferated trained academics in a large cancer center which might influenced results. In rural areas, these differences might be more pronounced.

Minor issue #2:

I would recommend to introduce sub-section in the methods parts to make it more reader-friendly.

Minor issue #3:

I would recommend to add all tables / figures at the end to make it more easy to access them.

6. PLOS authors have the option to publish the peer review history of their article (what does this mean?). If published, this will include your full peer review and any attached files.

Reviewer #1: **Yes: **Theodore M. Johnson II

Reviewer #2: **Yes: **Fabian Falkenbach

---

## [Author Response · Author response to Decision Letter 0]

3 Nov 2024

We have provided this as an attachment that will be easier to understand given the depth of response and revisions prompted by the Decision Letter.

---

## [Decision Letter · Decision Letter 1]

20 Nov 2024

Ascertaining provider-level implicit bias in electronic health records with rules-based natural language processing: a pilot study in the case of prostate cancer

PONE-D-24-33113R1

Dear Dr. Ramaswamy,

We’re pleased to inform you that your manuscript has been judged scientifically suitable for publication and will be formally accepted for publication once it meets all outstanding technical requirements.

Kind regards,

Alvaro Galli

Academic Editor

PLOS ONE

Additional Editor Comments (optional):

Reviewers' comments:

Reviewer's Responses to Questions

**Comments to the Author**

1. If the authors have adequately addressed your comments raised in a previous round of review and you feel that this manuscript is now acceptable for publication, you may indicate that here to bypass the “Comments to the Author” section, enter your conflict of interest statement in the “Confidential to Editor” section, and submit your "Accept" recommendation.

Reviewer #2: All comments have been addressed

2. Is the manuscript technically sound, and do the data support the conclusions?

Reviewer #2: Yes

3. Has the statistical analysis been performed appropriately and rigorously? 

Reviewer #2: Yes

4. Have the authors made all data underlying the findings in their manuscript fully available?

Reviewer #2: No

5. Is the manuscript presented in an intelligible fashion and written in standard English?

Reviewer #2: Yes

6. Review Comments to the Author

Reviewer #2: The authors have addressed all my comments in detail. The manuscript fulfills the quality criteria for publication.

7. PLOS authors have the option to publish the peer review history of their article (what does this mean?). If published, this will include your full peer review and any attached files.

Reviewer #2: **Yes: **Fabian Falkenbach

---

## [Editor Report · Acceptance letter]

13 Dec 2024

PONE-D-24-33113R1 

PLOS ONE

Dear Dr. Ramaswamy, 

I'm pleased to inform you that your manuscript has been deemed suitable for publication in PLOS ONE. Congratulations! Your manuscript is now being handed over to our production team.

Kind regards, 

on behalf of

Dr. Alvaro Galli 

Academic Editor

PLOS ONE